# Facts to Consider in Developing Materials That Emulate the Upper Jawbone: A Microarchitecture Study Showing Unique Characteristics at Four Different Sites

**DOI:** 10.3390/biomimetics8010115

**Published:** 2023-03-10

**Authors:** Ee Lian Lim, Wei Cheong Ngeow, Kathreena Kadir, Murali Naidu

**Affiliations:** 1Department of Oral and Maxillofacial Clinical Sciences, Faculty of Dentistry, University of Malaya, Kuala Lumpur 50603, Malaysia; elainelim@live.com.my (E.L.L.); kathreena@um.edu.my (K.K.); 2Department of Anatomy, Faculty of Medicine, University of Malaya, Kuala Lumpur 50603, Malaysia; murali_naidu@um.edu.my

**Keywords:** material, bone grafting, bone microarchitecture, maxilla, micro-computed tomography, trabecular bone

## Abstract

The maxilla is generally acknowledged as being more trabecular than the mandible. Allograft currently available for use in the maxillofacial region is harvested from the hip and long bones, irrespective of their local characteristics, and grafted onto the jawbones. Other alternative are autograft or commercially available bone substitutes. Due to their inherent differences, an in-depth understanding of the bone microarchitecture is important to develop the most compatible graft for use at the maxilla. This cross-sectional study aimed to determine the microstructures of bone harvested from different sites of the maxilla, to be used for standard setting. Forty-nine specimens from seven human cadavers were harvested from the zygomatic buttress, anterior maxillary sinus wall, anterior nasal spine and anterior palate. Each bone block, measuring of 10 mm × 5 mm, was harvested using rotary instruments. Bone analysis was performed following micro-computed tomography to obtain trabecular number (Tb.N), trabecular separation (Tb.Sp), trabecular thickness (Tb.Th), and bone volume fraction (BV/TV). There were site-related differences, with BV/TV that ranged between 37.38% and 85.83%. The Tb.N was the lowest at the palate (1.12 (mm^−1^)) and highest at the anterior maxillary sinus wall (1.41 (mm^−1^)) region. The palate, however, had the highest trabecular separation value (Tb.Sp) at 0.47 mm. The TB.Th was the lowest at the anterior nasal spine (0.34 mm) but both the zygoma and anterior maxillary sinus regions shared the highest Tb.Th (0.44 mm). Except for having the lowest Th.Sp (0.35 mm), the anterior maxillary sinus wall consistently showed higher values together with the zygomatic buttress in all other parameters. Concurring with current clinical practice of harvesting autograft from the zygomatic buttress and anterior maxillary sinus wall, their bony characteristic serve as the microarchitecture standard to adopt when developing new bone graft materials for use in the maxilla.

## 1. Introduction

Bone grafts are necessary to bridge bony maxillofacial defects caused by congenital deformity, pathology, trauma, and to restore bone resorption that occurs following tooth loss. Bone grafts and bone substitutes can be obtained from various sources using different techniques [1]. The gold standard is the transplantation of an autograft from another part of the patient’s body. Alternatives includes an allograft that is sourced from genetically non-identical members of the same species, a xenograft that is sourced from genetically non-identical members of different species and alloplastic materials, i.e., synthetic bone substitutes that negate the use of human or animal products. To date, developing an ideal substitute for human bone in the oral and maxillofacial region remains the holy grail of bone graft research [2,3]. Recent advances in biomaterial technology, such as digital light processing and partial infiltration, allows researchers to develop bioactive bone substitutes with the ability to control macroscopic pore size to 100–500 μm, with strong interconnected porosity [4], but the product is far from being clinically proven.

An autologous bone graft remains the gold standard because of its osteogenic, osteoinductive, and osteoconductive properties. Besides proper surgical technique, its microarchitecture is a crucial parameter for the success of bone grafting. In general, intramembranous bone is preferred than endochondral bone as it is subjected to less resorption, thus improving retention and long-term stability, while inducing the body to produce natural bone [5]. A calvarium graft despite being intramembranous of origin, may need to be performed under sedation or general anaesthesia, which increases cost with higher morbidity [6]. Therefore, more complex surgical procedures involving extraoral sites are the least preferred method for bone grafting in implant or periodontal therapy in which the volume of bone needed may be small.

Intraoral donor sites are locations most dentists and dental specialists are familiar with. These sites are easily reachable, and harvesting can be accomplished under local anaesthesia in a standard dental office setting. There are usually no scars left behind and intraoral donor sites have lower morbidity than extraoral sites [7]. Autografts from mandibular symphysis and ramus are frequently used for intraoral bone reconstruction [8], but these sites are located on the mandible. Some studies describe the use of autografts from the maxilla, especially the tuberosity. Other sites of interest are the zygomatic buttress [9,10], the anterior maxillary sinus wall [11,12], the anterior nasal spine [13,14], and the anterior palate [15,16]. However, the maxilla is generally acknowledged as being more trabecular than the mandible, although it also has the benefit of being of intramembranous of origin. The microarchitecture properties at various sites of the maxilla remain poorly documented despite its significance to autograft regeneration [17]. This may hinder our efforts to develop compatible synthetic bone materials that mimics the maxillary bone.

The term “bone quality” has been widely used in the literature to describe different aspects of bone characteristics. The trabecular bone microarchitecture is an essential factor determining bone quality [18], in addition to chemical composition. In fact, one study correlated the type I–IV of bone with various parameters used to determine bone quality [19]. An in-depth understanding of bone microarchitecture is important to develop the most compatible graft for use at the maxilla, so as to enhance treatment outcome [20].

In the past, conventional histomorphometry was widely used to investigate the micro-architectural properties of bone. As well as being an invasive procedure, this technique is limited to two-dimensional results and is time-consuming. In addition, conventional histomorphometry is destructive in nature, as the samples are destroyed in the process, preventing the specimens being studied repetitively [18]. With the advancement of micro-computed tomography (micro-CT), the three-dimensional microstructure of the human bone can now be clearly visualized and studied. This non-destructive and high-resolution method can depict the trabecular network in various shades of grey colour based on the mineral content. Many studies have routinely applied micro-CT in the structural evaluations of trabecular microstructure [19,21,22,23,24,25,26,27,28,29,30,31,32,33,34,35,36,37,38,39,40].

The current study aimed to determine the bone quantity and quality at four sites of interest, namely the zygomatic buttress, anterior maxillary sinus wall, anterior nasal spine and anterior palate; the first two of which have been recommend as donor sites in clinical practice. The objective was to obtain their trabecular number (Tb.N), trabecular separation (Tb.Sp), trabecular thickness (Tb.Th), bone volume fraction (BV/TV), and their correlation with structure model index (SMI) using micro-CT for standard setting. Findings from this study will provide much-needed information on the optimum microarchitecture of bone when developing biomaterial for use in grafting the maxilla.

## 2. Materials and Methods

This cross-sectional study was conducted at the Department of Anatomy, Faculty of Medicine, University of Malaya, with the micro-CT study undertaken at the Nuclear Agency Malaysia. The study received Institutional Board of Study approvals from the Faculty of Dentistry, University of Malaya [DF OS1814/0050(P)], and the Medical Research Ethics Committee, University of Malaya Medical Centre (MRECID.NO: 201865-6360). For the procurement of human bodies for the purpose of teaching and research in the university, consent was obtained based on the protocol adopted by the University of Malaya Medical Centre and approval by the Ministry of Health Malaysia. Seven human cadavers (all elderly males) previously embalmed and stored in 10% formalin solution (CH_2_O, 10%, Fisher Chemical) were obtained from the Department of Anatomy. Embalmed cadavers were stored between 6–12 months before use for research to ensure the tissues were fully fixed. The cadavers used in this study were thoroughly checked by the authors at the Department of Anatomy’s Dissection Hall where the cadavers were kept, to ensure that the jaw bones were intact and free from any past trauma, malformation or pathology. The absence of lesion(s), sign of previous surgery, or reconstructive procedures at the areas of investigation was confirmed. The data collection period commenced in October 2018 and ended in April 2019. The authors studied all cadavers available at that time and did not calculate the sample size. This is a limitation of the current study.

We investigated bone microarchitecture at the zygomatic buttress, anterior maxillary sinus wall, anterior palate, and anterior nasal spine (ANS), as shown in Figure 1. Each bone block, with a dimension of 10 mm × 5 mm (but different thickness due to difference in site of origin), was harvested using a fine (0.5 mm) round bur mounted on a rotary hand piece with the speed of 1200 rpm. A total of 49 bone blocks were obtained, with 42 of them derived from the first three sites of interest, while only one bone block was retrieved from the ANS region of each cadaver.

The sites of interest were identified using specific landmarks to ensure consistency. They were determined using the tooth size and dental arch dimension of southeast Asian Malay men as most of the cadavers were edentulous (Figure 1) [41]. The average distance from the central incisor to the second premolar was approximately 36 mm, hence the zygomatic buttress bone block was harvested at this distance bilaterally, 15 mm below the infraorbital rim. This location also corresponded with the hypothetical site of the bone graft harvesting site suggested by Lim et al., 2017 [42]. The average distance between central incisor and root apex of canines is approximately 20 mm, which is the reason the anterior maxillary sinus wall was identified as 20 mm from the midline. Bone blocks were harvested at 20 mm bilaterally. For the anterior palate, the bone blocks were harvested bilaterally at 2 mm from the bony crest, parallel to the tooth axis and 3 mm from the incisive foramen at the midline, following the description of Hassani et al. [43]. For the ANS region, only one bone block was harvested from each cadaver at 2 mm below its protrusion to avoid causing the collapse of the nose, based on descriptions from actual clinical practice.

The volume of the bone blocks harvested was measured using Archimedes’ principle [44] as this water displacement technique ensures that the air and spaces within them are not included. These bone blocks were then dried, preserved, and were subjected to micro-CT scan (SkyScan 1172, Kontich, Belgium). Only 21 bone specimens were sent for micro-CT scan due to cost constraints. They were fixed in a cylindrical shaped styrofoam before being fitted and mounted onto a holder to minimize movements during scanning. The micro-CT scanning parameters were set at 35 µm voxel size, 80 kVp, 124 mA, 0.5 mm aluminium filter, angular rotation step 0.70°, 180° scanning, with a total scan duration of 27 min and 43 s.

The micro-CT images were exported as TIFF files using NRecon (v 1.6, SkyScan, Kontich, Belgium) software for reconstruction and conversion into BMP images. Following this, they were imported into a trabecular bone analysis software, CTAn (v 1.15, SkyScan, Kontich, Belgium). The volume of interest (VOI) consisting of trabecular bone in each sample was delineated. The optimal threshold value for the images was determined using histogram analysis. Figure 2 shows a summary of the micro-CT bone analysis process.

Three-dimensional (3D) analysis was performed to acquire trabecular bone measurements, namely the bone volume fraction (BV/TV), trabecular number (Tb.N), trabecular thickness (Tb.Th), trabecular separation (Tb.Sp), and structure model index (SMI). The mean and standard deviation values were calculated using Microsoft Excel software (Version 2007, Microsoft, Redmond, WA, USA). The correlation of BV/TV, Tb.N, Tb.Th and Tb.Sp with structure model index (SMI) was determined. The trabecular bone parameters for the microarchitecture assessment are described in Table 1.

Statistical analyses were performed using SPSS Statistics 24.0 for Windows (SPSS, v.24.0, IBM, Chicago, IL, USA). Intra-observer reliability in measuring trabecular microstructure was performed using the intra-class correlation coefficient (ICC). Shapiro-Wilk test was done to verify the normality of data. Pearson’s correlation coefficient was used to assess the relationship between the corresponding measurement parameters. The statistical differences in the volume of bone harvested from various regions and their measurement parameters were evaluated by analysis of variance (ANOVA) based on a 5% confidence level.

## 3. Results

The mean volume of bone harvested ranged from 0.10 mL to 0.25 mL. Surprisingly, there was no significant difference in the mean volume of bone harvested from the zygomatic buttress (0.15 ± 0.05 mL), anterior maxillary sinus wall (0.16 ± 0.04 mL), anterior palate (0.14 ± 0.04 mL) and anterior nasal spine region (0.14 ± 0.05 mL), despite their differences in thickness.

Figure 3 shows the reconstructed three-dimensional appearance of bone scanned using micro-CT. Table 2 shows the three-dimension (3-D) microarchitecture evaluation of the maxillary bone measured at these four different sites of interest. The intra-observer reliability showed a 0.999 intra-class correlation coefficient (ICC) (CI 0.998 to 0.999; Value 762.853, df 202; *p* < 0.001), confirming good reliability of the measurements obtained. The average trabecular thickness (Tb.Th) was 0.42 ± 0.11 mm, with a range of 0.24 to 0.57 mm. Both the zygomatic buttress and anterior maxillary sinus wall had the highest mean trabecular thickness at 0.44 ± 0.14 mm and at 0.44 ± 0.09 mm, respectively. ANS had the lowest mean trabecular thickness at 0.34 ± 0.11 mm. Figure 4 presents the comparison of parameters between bone grafts taken from different donor sites.

One of the structure parameters, Tb.N, showed a value ranging from 0.71 to 1.86 mm^−^^1^ for the different anatomical sites. The lowest mean value was observed at the anterior palate, at 1.12 ± 0.21 mm^−^^1^ and the highest was at the anterior maxillary sinus wall (1.41 ± 0.36 mm^−^^1^). The anterior palate also had the lowest BV/TV (44.79 ± 4.43%), while the anterior maxillary sinus wall had the highest BV/TV (60.11 ± 9.30%). This contrasts with results observed for the Tb.Sp, which ranged from 0.22 to 0.92 mm. The anterior palate had the highest value of trabecular separation (0.47 ± 0.06 mm) while the anterior maxillary sinus wall had the lowest value (0.35 ± 0.10 mm). All together, these findings demonstrated that the anterior palate had the highest trabecular separation with the lowest trabecular number, and the lowest BV/TV.

All parameters showed no significant difference between different sites, except for structural model index (SMI) (F (3,17) = 3.248, *p* = 0.048), when analysed using one-way ANOVA. The post-hoc Dunnett T3 test revealed that the anterior maxillary sinus wall had a significantly lower SMI (−0.80 ± 0.83) compared to the anterior palate (0.80 ± 0.31). Similarly, ZYGOMA showed a significantly lower SMI (−2.51 ± 3.60) compared to the anterior palate (0.80 ± 0.31). In short, the anterior palate had the highest SMI.

Lastly, correlation analysis was done between different bone quality parameters [19,29,31]. BV/TV had a strong inverse correlation with trabecular separation (Tb.Sp) (r = −0.673, *p* = 0.001). The high negative correlation between BV/TV and Tsp. showed values of BV/TV higher than 50% squares with bones presenting most of their trabeculae separation less than 0.40 mm between each sample. BV/TV showed a moderate correlation with trabecular thickness (Tb. Th) (r = 0.547, *p* = 0.01). The positive correlation coefficient between BV/TV and Tb.Th shows a value of BV/TV higher than 50% squares, with bones presenting most of their trabeculae thicker than 0.40 mm.

SMI represents a measurement of surface convex curvature. There was a strong inverse correlation between SMI and BV/TV (r = −0.882, *p* < 0.001). In this study, 90.9% of bone specimens with BV/TV values lower than 50% showed a positive value of SMI, with a preponderance of values between 0 and 1 (plate-like trabeculae). Almost forty percent (38.10%) of bone specimens showed negative bone values, while 61.9% of bone specimens showed positive values. Only 4.76% of the bone had positive values between 1 and 2, with equal preponderance of plate-like and rod-like trabeculae.

SMI had a moderate inverse correlation with trabecular number (Tb. N) (r = −0.471, *p* = 0.031) and trabecular thickness (Tb. Th) (r = −0.414, *p* = 0.062). A poor direct correlation was recorded between trabecular separation (Tb. Sp) and trabecular thickness (Tb. Th) (r = 0.126, *p* = 0.587). Correlation studies help to identify the absence or presence of a relationship between the trabecular bone parameters, but our findings suggest that there is no straightforward relationship within the parameters used in the analysis. This is possibly undermined by the unique characteristics at the four different sites as described above.

In summary, the microarchitecture demonstrated a high value of bone volume fraction, indicating good bone quality from the four sites of interest. Moreover, the high value of trabecular number and the normal mean value of trabecular separation indicated good compact structure at all four sites.

## 4. Discussion

This study was performed on embalmed cadavers because of the difficulty in obtaining fresh human cadavers. Due to our tropical climate, which is hot and humid, fresh cadavers can undergo deterioration rapidly. Therefore, the cadavers were formalin-fixed as soon as possible after death [46]. Holm and Iazzo [47] highlighted that formalin fixation is still the most widely used and cost-effective technique for embalming human cadavers for education and medical research. Although there is evidence that formalin significantly alters the mechanical/biomechanical behaviour of ligaments, spine, and bone of animals, the histologic structure, density, and the mineral and lipid parameters do not seem to be affected [48]. Thus, this preservation method serves the purpose of the current study.

One of the limitations of this study was the lack of individual information on the donors’ age and medical history. Thus, it was impossible to determine if the microarchitecture of bone blocks studied were affected by age or medical disorders that could alter bone quantity and quality.

Table 3 summarizes the microarchitecture at various human bone sites. Several studies have reported that the BV/TV ratio was the most significant parameter in determining bone quality [32]. Increased BV/TV was found in areas where great masticatory forces were received, as a result of optimal bone adaptation to the masticatory forces [32]. A greater value of BV/TV indicates a more compact structure that is able to resist the external forces transferred via mastication. The bone volume fraction (BV/TV) of the maxilla ranged from 13.53% to 48.70%, while that in the mandible ranged from 9.24 to 43.74%, depending on the site of study (see Table 3). The mean value of the BV/TV (53.04 ± 14.31% (range: 37.38–85.53%)) found in the present study was higher than other similar studies. Kim and Henkin [30], for example, reported the BV/TV values were 14.59 ± 7.68% for maxilla and 27.28 ± 10.19% for mandible with these values ranging from 6.75% to 48.92%. Parsa et al. attributed this difference to the difference in samples, ROI selections and the scanner system used [32].

The trabecular bone number (Tb.N), a parameter derived from the endosteal space architecture, is a good indicator of bone quality. In general, a high trabecular number suggests a more compact structure. The Tb.N of maxillary bone has been reported to be between 0.62 to 2.19 mm^−1^, while values in the mandible range between 0.007 and 2.19 mm^−1^ (see Table 3). The present study reported a higher mean value that, nevertheless, falls within the range reported in the literature. However, the range of trabecular number reported in the literature is inconsistent, and in general were found to be higher in the Caucasians. One study reported that the cadaveric maxillary posterior regions had greater trabecular number (1.83 ± 0.25 mm^−1^) [36], while Kim and Henkin [30] reported greater values from the maxilla (2.07 ± 0.80 mm^−1^) and mandible (3.76 ± 1.99 mm^−1^). These differences could be attributed to bigger and denser bones among the Caucasians compared to the Asian population.

The trabecular thickness (Tb.Th) of the maxillary bone has been reported to be between 0.10 to 0.50 mm, while values in the mandible range between 0.004 and 0.31 mm (see Table 3). In comparison to natural bone, the Tb.Th has been reported to be significantly thinner in the areas augmented with an allograft [50]. Values reported in the current study lean towards the higher end of the range, and are higher than the figures reported for the mandible.

Trabecular separation (Tb.Sp) is another indicator of bone quality. The trabecular separation (Tb.Sp) of maxillary bone has been reported to be between 0.31 to 0.89 mm, while values in the mandible has a range between 0.007 and 1.31 mm (see Table 3). By comparison, the value of trabecular separation found in the present study was lower than the D1 bone reported by Lee et al. [19], but had similar values to those reported by Kim and Henkin [30] for the mandibular region (0.42 ± 0.18 mm). As the trabecular separation reported here leaned towards D1 bone values, it can be concluded that the trabeculae were in close proximity, forming a compact trabecular bone structure. It is interesting that other authors reported greater trabecular separation value in the maxilla region (0.63 ± 0.18 mm) [28] and maxillary posterior region (0.57 ± 0.13 mm) [36], suggestive of variability in human jaw bone quality. The Tb.Sp value found in the present study is within the range of the optimal pore size for bone ingrowth (150–600 μm) [51]. This pore size is associated with better osteoconductivity, another important factor that cannot be overlooked when developing new biomaterials.

The parameter of SMI was described by Hildebrand & Rüegsegger (1997) to determine the 3D morphology of trabecular bone [52]. SMI represents a measurement of surface convex curvature by dilation of the 3D voxel model, that is, artificially adding one voxel thickness to all binarised object surfaces [52]. SMI is derived as follows:$$SMI=6\times \left(\frac{S^{\prime }\times V}{S^{2}}\right)$$
where *S* is the object surface area before dilation and *S′* is the change in surface area caused dilation. *V* is the initial, undilated object volume.

This parameter is crucial in osteoporotic degradation of trabecular bone that is characterized by the presence of rod- and plate-like structures in various configurations [53]. An ideal plate, cylinder and sphere have SMI values of 0, 3 and 4 respectively. In contrast, cylindrical and spherical cavities have SMI of −3 and −4 respectively [54]. The concave surfaces of closed cavities demonstrate the negative convexity for the calculation of SMI. Hence, bony regions containing greater numbers of enclosed cavities (usually regions with BV/TV >50%) may have negative SMI values. In the current study, the mean value of SMI is −0.62 ± 2.33, with a range of −7.11 to 1.27. The negative value of SMI seen in the zygomatic buttress (−2.51 ± 3.60) and anterior maxillary sinus wall (−0.80 ± 0.83) can be explained by the fact that these bones are solid and have a compact structure with a percent volume (BV/TV) above 50%. Negative convexity to the SMI parameter was found in 38.10% of bone specimens among our series, indicating their origin from a denser bone [29,54]. Only 4.76% of the bone had positive values between 1 and 2, with equal preponderance of plate-like and rod-like trabeculae.

Bone quality at both the donor and recipient sites dictates treatment outcome in human donor autograft procedures. However, bone quality is independent of bone mass, as observed in bone with pathological disorder [55]. Therefore, the alternative is to use allograft, xenograft or synthetic bone material [1]. However, allografts currently available for use in dentistry are harvested from the hip and long bones and are grafted onto the jawbones irrespective of their local characteristics. The micro-architecture of trabecular bone has been proven to be among the crucial elements of bone quality, as it determines the bone healing rate. However, emulating this in synthetic material remains problematic.

Apart from particulate autogenous bone, demineralised freeze-dried bone allografts, deproteinized bovine bone allografts, synthetic β-tricalcium phosphate (β-TCP), anhydrous dicalcium phosphate (monetite), synthetic biphasic calcium phosphate (BCP), magnesium-enriched bioceramic, coralline-derived biomaterials, and porous titanium granules have been studied clinically, with their microarchitecture analysed using micro-CT [49] However, all these bone graft materials have been shown to differ from the normal microarchitecture of the recipient sites in the maxilla and mandible in various microarchitecture parameters.

Therefore, the three-dimensional evaluation of human cadaver maxillary bone microarchitecture in this study allows for better understanding of the qualitative differences between the trabecular parameters among different sites, allowing researchers to appreciate their unique characteristics that differ from the mandible [56]. Concurring the current clinical practice of harvesting autografts from the zygomatic buttress and anterior maxillary sinus wall, their bony characteristic serve as the microarchitecture standard to adopt when developing new bone graft materials for use in the maxilla.

## 5. Conclusions

The quantity and quality of bone harvested from the zygomatic buttress, anterior maxillary sinus wall, anterior palate and anterior nasal spine were determined by a water displacement technique and a micro-CT scan, respectively. There was no significant difference in all bone parameters, with the exception of the SMI. Each site has its unique combinations of parameters. The BV/TV ranged between 37.38% and 85.83%, with anterior maxillary sinus wall > zygoma > palate > anterior nasal spine. The high values of bone volume fraction indicated good bone quality at the four sites of interest. The high value of trabecular number and the normal mean value of trabecular separation indicated good compact structure at all four sites. Except for having the lowest Th.Sp (0.34 mm), the anterior maxillary sinus wall consistently showed higher values together with the zygomatic buttress in all other parameters. Concurring with the current clinical practice of harvesting autografts from the zygomatic buttress and anterior maxillary sinus wall, their bony characteristic serve as the microarchitecture standard to adopt when developing new bone graft materials for use in the maxilla.

## Figures and Tables

**Figure 1 biomimetics-08-00115-f001:**
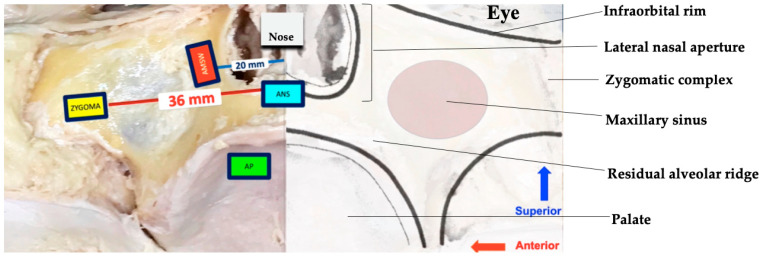
(**Left**) The four sites of interest in this study were the zygomatic buttress (ZYGOMA), anterior maxillary sinus wall (AMSW), anterior palate (AP), and anterior nasal spine (ANS). The location of the bone block was determined by using dental morphometrics. The average distance between the central incisor and the second premolar was approximately 36 mm, based on the mesiodistal sum teeth concerned. The average distance between central incisor and root apex of canine is approximately 20 mm. (**Right**) Anatomical scheme of the midface.

**Figure 2 biomimetics-08-00115-f002:**
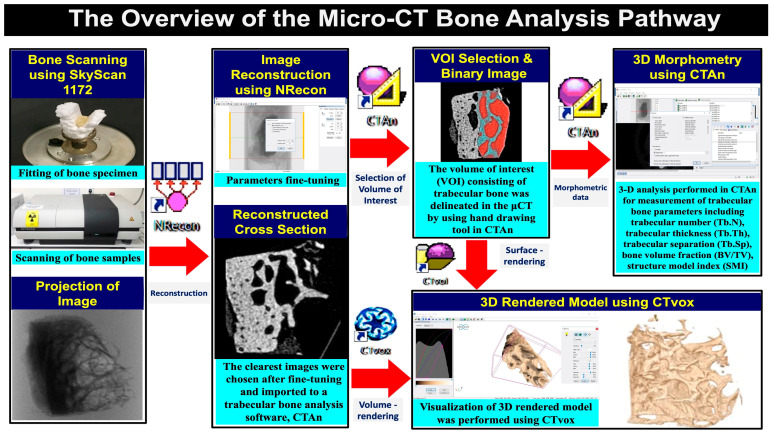
An overview of the micro-CT bone analysis process. All the steps were supported by SkyScan software. The steps for the micro-CT bone analysis pathway were bone scanning using SkyScan 1172, image reconstruction using NRecon, the volume of interest (VOI) selection and 3D morphometry acquisitions using CTAn. Visualization of the 3D rendered model was performed using CTvox or CTvol.

**Figure 3 biomimetics-08-00115-f003:**
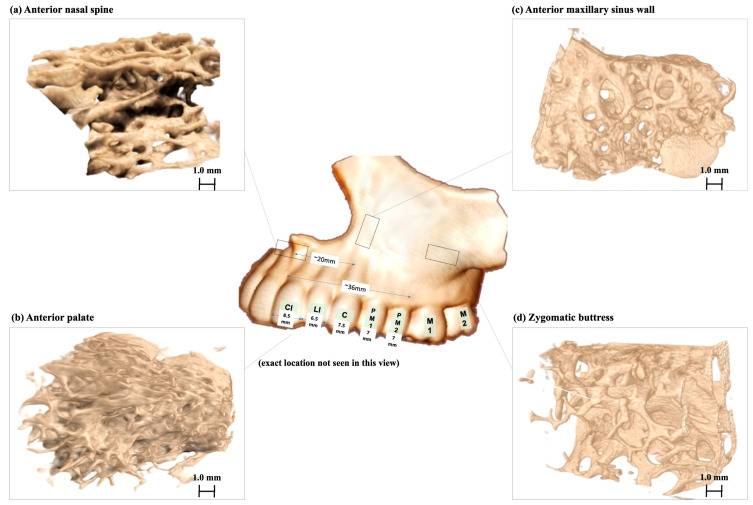
The reconstructed appearance of bone specimens visualised using CTvox software. (**a**) Anterior nasal spine, (**b**) anterior palate, (**c**) anterior maxillary sinus wall, (**d**) zygomatic buttress.

**Figure 4 biomimetics-08-00115-f004:**
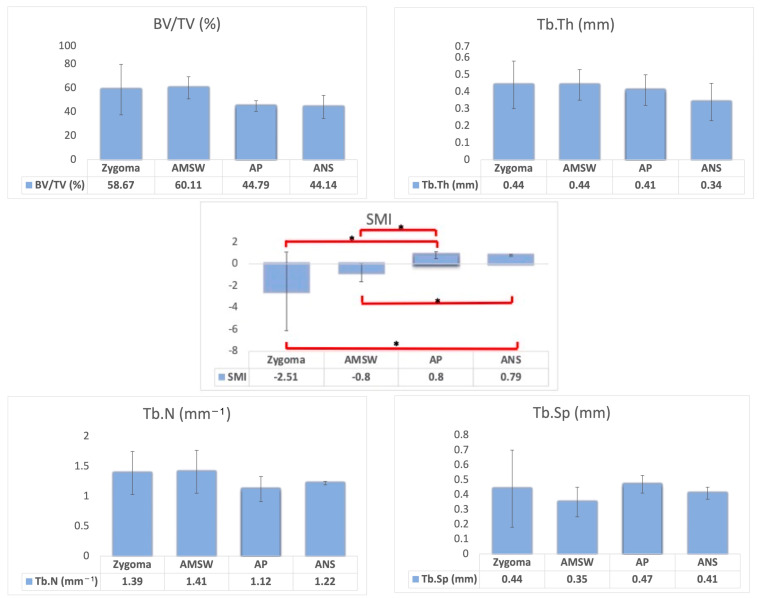
Comparison of parameters between bone grafts taken from different donor sites. ***** ANOVA; *p* < 0.05. Note: The sites of interest in this study are the zygomatic buttress (ZYGOMA), anterior maxillary sinus wall (AMSW), anterior palate (AP), and anterior nasal spine (ANS).

**Table 1 biomimetics-08-00115-t001:** Trabecular bone parameters for the microarchitecture assessment [45].

Abbreviation	Variable	Description	Standard Unit
TV	Total volume	Volume of the entire region of interest	mm^3^
BV	Bone volume	Volume of the region segmented as bone	mm^3^
BV/TV	Bone volume fraction	Ratio of the segmented bone volume to the total volume of the region of interest	%
Tb.N	Trabecular number	Measure of the average number of trabeculae per unit length	mm^−1^
Tb.Th	Trabecular thickness	Mean thickness of trabeculae, assessed using direct 3D methods	mm
Tb.Sp	Trabecular separation	Mean distance between trabeculae, assessed using direct 3D methods	mm
SMI	Structure model index	An indicator of the structure of trabeculae	none

**Table 2 biomimetics-08-00115-t002:** Comparison of parameters between bone grafts taken from different donor sites (*n* = 21).

Donor Site	Mean ± SD	Minimum	Maximum	F	*p*-Value
Tissue Volume (mm^3^)				1.310	0.304
ZYGOMA	102.07 ± 52.83	27.21	199.57		
AMSW	76.31 ± 31.92	49.09	135.50		
AP	83.56 ± 26.79	51.09	113.23		
ANS	50.96 ± 24.63	27.21	76.38		
Overall	82.1 ± 38.32	27.21	199.57		
Bone Volume (mm^3^)				2.959	0.062
ZYGOMA	58.98 ± 29.17	23.25	93.17		
AMSW	44.31 ± 13.48	29.23	64.21		
AP	37.37 ± 12.09	23.27	51.47		
ANS	21.29 ± 8.08	15.03	30.40		
Overall	43.23 ± 21.41	15.03	93.17		
Bone Volume/Tissue Volume (%)				2.179	0.128
ZYGOMA	58.67 ± 21.16	38.59	85.83		
AMSW	60.11 ± 9.30	47.38	70.01		
AP	44.79 ± 4.43	38.38	52.08		
ANS	44.14 ± 9.68	37.38	55.23		
Overall	53.04 ±14.31	37.38	85.83		
Structural Model Index				3.248	0.048 *
ZYGOMA	−2.51 ± 3.60	−7.11	0.46		
AMSW	−0.80 ± 0.83	−1.47	0.52		
AP	0.80 ± 0.31	0.46	1.27		
ANS	0.79 ± 0.14	0.63	0.87		
Overall	−0.63 ± 2.33	−7.11	1.27		
Trabecular Thickness (mm)					
ZYGOMA	0.44 ± 0.14	0.25	0.57	0.669	0.583
AMSW	0.44 ± 0.09	0.33	0.56		
AP	0.41 ± 0.09	0.27	0.53		
ANS	0.34 ± 0.11	0.24	0.45		
Overall	0.42 ± 0.11	0.24	0.57		
Trabecular Number (mm^−1^)				1.238	0.327
ZYGOMA	1.39 ± 0.36	0.71	1.74		
AMSW	1.41 ± 0.36	1.03	1.86		
AP	1.12 ± 0.21	0.86	1.40		
ANS	1.22 ± 0.03	1.19	1.23		
Overall	1.29 ± 0.30	0.71	1.86		
Trabecular Separation (mm)				0.692	0.570
ZYGOMA	0.44 ± 0.26	0.22	0.92		
AMSW	0.35 ± 0.10	0.23	0.50		
AP	0.47 ± 0.06	0.39	0.55		
ANS	0.41 ± 0.04	0.39	0.46		
Overall	0.42 ± 0.15	0.22	0.92		

* Significant at *p*-value less than 0.05. F = Value of distribution, generated by dividing two mean squares to determine test significance. Note: the sites of interest in this study are the zygomatic buttress (ZYGOMA), anterior maxillary sinus wall (AMSW), anterior palate (AP), and anterior nasal spine (ANS).

**Table 3 biomimetics-08-00115-t003:** Summary of the bony microarchitecture of various sites.

Authors (Year)	Population	Mean Age (Range)	n */N **	Method	Site(s)	Variable (If Any)	BV/TV (%)	Tb.N (1/mm)	Tb.Th (mm)	Tb.Sp (mm)
Muller et al., 1998 [21]	American	68 ± 16 (23–92)	70	micro-CT	Transiliac bone		14.48 ± 5.34	-	0.11 ± 0.02	0.77 ± 0.35
Giesen & Van Eijden, 2000 [22]	Dutch	72.6 ± 11.2 (56–89)	99 (11)	micro-CT	Condyles		17.00 ± 5.00	1.66 ± 0.26	0.10 ± 0.02	0.52 ± 0.13
Moon et al., 2004 [23]	Korean	55.1 (29–75)	10	micro-CT	Mandible (Alveolar Bone)		43.74 ± 16.04	1.27 ± 0.24	0.31 ± 0.08	0.51 ± 0.14
					Mandible (Basal Bone superior to mandibular canal)		20.39 ± 6.45	0.90 ± 0.23	0.28 ± 0.09	0.88 ± 0.20
					Mandible (Basal Bone inferior to mandibular canal)		9.24 ± 7.11	0.70 ± 0.20	0.22 ± 0.05	1.31 ± 0.42
Kato et al., 2005 [24]	Japanese	79.6	56 (28)	micro-CT	Jugale		23.2 ± 4.3	0.16 ± 0.05	1.53 ± 0.48	0.56 ± 0.20
					Middle point		19.9 ± 5.4	0.15 ± 0.05	1.38 ± 0.33	0.62 ± 0.28
					Zygomaxillae		20.5 ± 6.5	0.15 ± 0.06	1.49 ± 0.40	0.58 ± 0.20
Siddiqi et al., 2013 [25]	New Zealander	80 (65–94)	16	micro-CT	Median Palate	Palate	42.9 ± 13.8	1.1 ± 0.3	0.4 ± 0.2	7.5 ± 4.7
					Maxillary Premolar	Premolar	38.1 ± 12.5	1.0 ± 0.6	0.5 ± 0.3	8.1 ± 5.6
Ulm et al., 2009 [26]	Austrian	77.58 ± 10.09	278 (128)	micro-CT	Mandible (lateral incisor)	Female	30.70 ± 9.91	1.50 ± 0.34	0.19 ± 0.05	0.46 ± 0.14
						Male	36.90 ± 12.40	1.77 ± 0.39	0.21 ± 0.06	0.38 ± 0.14
					Mandible (first premolar)	Female	24.50 ± 8.45	1.47 ± 0.43	0.17 ± 0.04	0.57 ± 0.20
						Male	35.90 ± 13.62	1.58 ± 0.32	0.22 ± 0.06	0.82 ± 0.27
					Mandible (first molar)	Female	20.90 ± 9.65	1.22 ± 0.37	0.17 ± 0.04	0.72 ± 0.28
						Male	24.50 ± 7.93	1.38 ± 0.30	0.17 ± 0.04	0.58 ± 0.18
Blok et al., 2012 [27]	Dutch	73.7 ± 12.5	10	micro-CT	Maxilla		24.0 ± 13.0	1.57 ± 0.56	0.20 ± 0.05	0.69 ± 0.24
					Mandible		37.0 ± 18.0	1.50 ± 0.42	0.29 ± 0.11	0.71 ± 0.25
Kim et al., 2013 [28]	Korean	NA	69 (4)	micro-CT	Anterior Maxilla		21.35 ± 5.18	0.99 ± 0.23	0.22 ± 0.05	0.72 ± 0.16
					Posterior Maxilla		17.68 ± 6.21	0.89 ± 0.27	0.20 ± 0.07	0.79 ± 0.14
					Anterior Mandible		23.87 ± 7.68	0.72 ± 0.312	0.33 ± 0.05	0.85 ± 0.13
					Posterior Mandible		18.46 ± 9.44	0.78 ± 0.26	0.23 ± 0.07	0.82 ± 0.27
González-García & Monje, 2013 [29]	Spanish	51.56 ± 13.78 (20–79)	52 (31)	micro-CT	Maxilla		48.70 ± 17.85	2.19 ± 0.71	0.22 ± 0.06	0.31 ± 0.10
Kim & Henkin, 2015 [30]	American	NA	34 (12)	micro-CT	Maxilla		14.59 ± 7.68	2.07 ± 0.80	0.10 ± 0.02	0.63 ± 0.18
					Mandible		27.28 ± 10.19	3.76 ± 1.99	0.09 ± 0.02	0.42 ± 0.18
Bertl et al., 2015 [31]	Austrian	NA	36 (12)	micro-CT	Anterior Maxilla		27.15 ± 7.90	1.051 ± 0.20	0.26 ± 0.04	0.59 ± 0.13
					Posterior Maxilla		13.54 ± 3.40	0.624 ± 0.14	0.22 ± 0.03	0.89 ± 0.14
					Zygoma		26.79 ± 7.40	1.024 ± 0.20	0.26 ± 0.04	0.63 ± 0.13
Parsa et al., 2015 [32]	Dutch	NA	20	micro-CT	Mandible	micro-CT	32.35 ± 18.81	-	-	-
				CBCT		CBCT	36.79 ± 23.17	-	-	-
Kim et al., 2015 [33]	Korean	NA	68 (4)	micro-CT	Maxilla	Imaging Protocol				
					Mandible	19.37 µm	18.53 ± 8.17	0.24 ± 0.07	0.77 ± 0.27	0.83 ± 0.17
						96.87 µm	18.15 ± 8.60	0.38 ± 0.11	0.47 ± 0.17	0.95 ± 0.19
Lee et al., 2017 [19]	Korean	75.7	116	micro-CT	Maxilla	Bone Density				
		(67–96)	(30)		Mandible	D1	37.29 ± 17.96	1.21 ± 0.45	0.30 ± 0.08	0.59 ± 0.22
						D2	27.46 ± 9.58	0.99 ± 0.24	0.28 ± 0.06	0.68 ± 0.14
						D3	18.40 ± 10.20	0.71 ± 0.26	0.25 ± 0.05	0.82 ± 0.19
						D4	9.83 ± 8.02	0.41± 0.27	0.22 ± 0.06	1.20 ± 0.48
Suttapreyasri et al., 2018 [35]	Thailand	>20	62 (41)	micro-CT	Maxilla	Location				
				CBCT	Mandible	Anterior Maxilla	35.23 ± 10.68	-	-	-
						Posterior Maxilla	36.11 ± 9.15	-	-	-
						Anterior Mandible	63.25 ± 19.85	-	-	-
						Posterior Mandible	46.74 ± 13.14	-	-	-
Kulah et al., 2019 [36]	Turkish	NA	17	micro-CT	Maxilla		32.65 ± 7.46	1.83 ± 0.05	0.28 ± 0.05	0.57 ± 0.13
Kivovics et al., 2020 [49]	Hungarian	54.7 ± 6.5	16 (9)	micro-CT	Maxilla * Augmented sinus	micro-CT	12.25	-	0.15	0.88
				CBCT	(grafted with allograft)	CBCT	81.29	-	1.82	0.85
Ibrahim et al., 2021 [37]	Dutch	NA	25	micro-CT	Mandible	micro-CT				
						Anterior Posterior	0.008 ± 0.003 0.007 ± 0.004	0.005 ± 0.008 0.004 ± 0.001	0.007 ± 0.008 0.009 ± 0.001	- -
				CBCT		CBCT				
						Anterior Posterior	0.006 ± 0.002 0.005 ± 0.003	0.007 ± 0.002 0.006 ± 0.002	0.009 ± 0.003 0.010 ± 0.004	- -
Tabassum et al., 2022 [38]	Malaysian	26.6 ± 5.9 (22–43)	20	CBCT	Mandible	CBCT	44.40 ± 14.77	0.44 ± 0.15	1.25 ± 0.55	2.05 ± 0.75
Tayman et al., 2022 [39]	Turkey	NA	12	micro-CT CBCT	Posterior mandible	Micro CT (Std) Micro CT (Hi) CBCT (Std) CBCT (Hi)	46.01 ± 8.48 44.28 ± 8.47 57.13 ± 11.10 54.45 ± 11.98	2.05 ± 0.46 2.04 ± 0.47 1.43 ± 0.25 1.43 ± 0.29	0.24 ± 0.06 0.23 ± 0.06 0.46 ± 0.09 0.44 ± 0.09	0.53 ± 0.11 0.51 ± 0.11 0.48 ± 0.12 0.46 ± 0.12
El-Gizawy et al., 2023 [40]	USA	21	4 (1)	micro-CT	Distal femoral condyle		21.01 ± 4.72	-	1.25 ± 0.55	0.35 ± 0.03
Current study	Malaysian	Elderly	49 (7)	micro-CT	Maxilla	Location				
						Zygoma	58.67 ± 21.16	1.39 ± 0.36	0.44 ± 0.14	0.44 ± 0.26
						AMSW	60.11 ± 9.30	1.41 ± 0.36	0.44 ± 0.09	0.35 ± 0.10
						AP	44.79 ± 4.43	1.12 ± 0.21	0.41 ± 0.09	0.47 ± 0.06
						ANS	44.14 ± 9.68	1.22 ± 0.03	0.34 ± 0.11	0.41 ± 0.04
						Mean	53.04 ±14.31	1.29 ± 0.30	0.42 ± 0.11	0.42 ± 0.15

* n is the number of specimens. ** N is the number of cadavers. CBCT = cone-beam computed tomography. NA = data (age) not available. Note: The sites of interest in this study are the zygomatic buttress (ZYGOMA), anterior maxillary sinus wall (AMSW), anterior palate (AP), and the anterior nasal spine (ANS).

## Data Availability

No new data were created or analyzed in this study. Data sharing is not applicable to this article.

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
