# Peer review of "Facts to Consider in Developing Materials That Emulate the Upper Jawbone: A Microarchitecture Study Showing Unique Characteristics at Four Different Sites"

_biomimetics, 2023, doi:10.3390/biomimetics8010115_

Round 1
Reviewer 1 Report
Reviewer’s Comments:
The manuscript “Challenges in Making Materials that Emulate the Upper Jaw Bone - Anatomical Study Shows Differences at Different Sites” is very interesting work. In this work, the maxilla is generally acknowledged as being more trabecular than the mandible. Allograft currently available for use in dentistry is harvested from the hip and long bones, and grafted onto the jawbones irrespective of their local characteristics. Thus, an in-depth understanding of the bone microarchitecture is important to develop the most compatible graft for use at the maxilla. This cross-sectional study aimed to determine the microstructures of bone harvested from different sites of the maxilla, to be used for standard setting. Forty-nine specimens from seven human cadavers were harvested from the zygomatic buttress, anterior maxillary sinus wall, anterior nasal spine, and anterior palate. Each bone block measuring of 10mm x 5mm was harvested using rotary instruments. However, the following issues should be carefully treated before publication.
1. In abstract, the author should add more scientific findings.
2. Keywords: the synthesized system is missing in the keywords. So, modify the keywords.
3. In the introduction part, the introduction part is not well organized and cited references should cite recently published articles such as 10.3390/molecules27217368, 10.3389/fchem.2022.1023316
4. Introduction part is not impressive and systematic. In the introduction part, the authors should elaborate on the scientific issues in Anatomical research.
5. Results …, The author should provide reason about this statement “The lowest mean value was observed at the anterior palate (1.12) 193 and the highest was demonstrated at the anterior maxillary sinus wall (1.41)”.
6. The authors should explain regarding the recent literature why “Correlations analyses were done between different bone quality parameters”.
7. Material and method: Materials “write all the detail of chemicals in unique format rather than to write individual chemical such as Tin Chloride (SnCl4.5H2O). It should be written as “tin chloride (SnCl4.5H2O, 98%, Sigma)”. Write all the chemicals in this format.
8. The authors should explain regarding the recent literature why “This study was performed on embalmed cadavers because of the difficulty to obtain fresh human cadavers; they also undergo rapid deterioration and carry potential risk of infection”.
9. Comparison of the present results with other similar findings in the literature should be discussed in more detail. This is necessary in order to place this work together with other work in the field and to give more credibility to the present results.
10. The conclusion part is very week. Improve by adding the results of your studies.
Reviewer 2 Report
The authors have addressed an important research question, which will certainly assist clinicians as well as researchers. However, I have a few suggestions/comments for the authors which should be considered:
1. The title of the paper needs to be revised as in the current form it reflects that authors have researched on the challenges in making materials which in fact they haven't. It is suggested that the title should be written in line with the main objective of study.
2. The information in the introduction section should be supported with suitable references. For instance, in lines 86 and 87, the authors have mentioned "Many studies have routinely applied µ-CT in the structural evaluations of trabecular microstructure [9]" and they have given only one reference.
3. It is suggested that the authors should add a clear objective of the research work along with the hypothesis.
4. How did the authors calculate the sample size?
5. How did the authors thoroughly check the cadavers [Lines 105 and 106]?
6. The authors should mention the average storage time of cadavers since their death [Lies 103 and 104].
7. What was the speed of round bur whilst harvesting each bone block? It would be better for the readers if the authors would add the images of experimental work .
8. Many references are very old, therefore, it is suggested that the authors should update the references.
Reviewer 3 Report
This study, which aimed to determine the microstructures of bone harvested from different jaw sites, is very attractive to be used for the standard setting or for data comparison.
This study is relevant as it will increase the understanding on the internal structure of some bone sites and at the same time explain the meaning of the morphometric parameters in relation to their obtained value.
The only question I would like to ask the authors is the following:
"Why did you use the water displacement technique for the calculation of the sample volume and not the analysis software CT-An, which was also used in this study?"
Thank you.
Reviewer 4 Report
The authors investigate the differences in bone microstructure in the human jaw. They imaged a lot of bone parts and extracted bone parameters from µCT images. However, a proper discussion and analysis is missing, in terms of obtained values (example of SMI values not discussed) or differences observed with literature (example of the BV/TV valules). The authors basically compare their values to literature, without necessarily understanding what the values represent (negative SMI is aberrant) and not comparing them in a correct statistic manner (comparison of a group to one mean value). The mostly referred study, from Kim et al (2013) is basically the same, but better conducted. This study does not add any particularly interesting nor enough new results to be valid for publication.
Detailed comments:
The mismatch between the title “Anatomical Study Shows Differences at Different Sites” and the conclusions “there was no significant difference” is surprising...
Typos. The paper presents grammatical errors or typos, and needs to be re-read. Eg L48 “inducing *to* body”, L55 “implant graft therapy *when* the volume”, L150 “35*mm* voxel size”, L233 “trabecular number (*Tb.Th*), L314 “literatures”, etc.
Some terms would need to be defined. Such as the SMI (L187), eg how it’s calculated and what it represents. Negative values (aberrant) are not explained nor discussed. Moreover, the bone imaging community usually state that the “SMI is of no use to quantify real bone structures”. Also, the concepts of “test values” (L210, what is it? units??) and “D1” (L321) are not clear and should be better explained.
Figures: they are relatively useless as now. Fig 1 would need a general view of the jaw, potentially merging with Fig 2, to clearly identify the harvesting spots. Some uCT results would also be needed to attest of the image quality, visual differences, better explanation of the image processing pipeline, etc.
Statistics and comparison with literature. The authors compare their results with the values extracted from the study from Kim et al. (2013), but they base all their comparison on one value, without taking the whole range of values obtained. For example, the BV/TV value of “23.87” (Table 3) is used as a single value, when the SD in the original article is around 12 ; this is strongly biased. Moreover, the authors note differences with literature but do not (even try to) give an explanation (L307).
Round 2
Reviewer 4 Report
Please find my comments in the enclosed pdf document
